# Measuring Sparse Autoencoder Feature Sensitivity

## Abstract

Sparse Autoencoder (SAE) features have become essential tools for mechanistic interpretability research. SAE features are typically characterized by examining their activating examples, which are often "monosemantic" and align with human interpretable concepts. However, these examples don't reveal *feature sensitivity*: how reliably a feature activates on texts similar to its activating examples. In this work, we develop a scalable method to evaluate feature sensitivity. Our approach avoids the need to generate natural language descriptions for features; instead we use language models to generate text with the same semantic properties as a feature's activating examples. We then test whether the feature activates on these generated texts. We demonstrate that sensitivity measures a new facet of feature quality and find that many interpretable features have poor sensitivity. Human evaluation confirms that when features fail to activate on our generated text, that text genuinely resembles the original activating examples. Lastly, we study feature sensitivity at the SAE level and observe that average feature sensitivity declines with increasing SAE width across 7 SAE variants. Our work establishes feature sensitivity as a new dimension for evaluating both individual features and SAE architectures.

## 1 Introduction

Sparse Autoencoders (SAEs) have emerged as a powerful technique to identify meaningful directions in language model activation spaces (Cunningham et al., 2023; Templeton et al., 2024). These learned directions, or SAE features, have proven to be valuable for mechanistic interpretability. Use cases include: surfacing surprising information present in model activations (Templeton et al., 2024; Ferrando et al., 2025), controlling model behavior via activation steering (Durmus et al., 2024; Nanda et al., 2025), identifying computational circuits within models (Ameisen et al., 2025; Marks et al., 2025a; Lindsey et al., 2025), and more open-ended exploration of training data (Marks et al., 2025b) or other datasets (Movva et al., 2025; Jiang et al., 2025).

A key step in almost all SAE applications is to first characterize each SAE feature. This is commonly done by examining example inputs that activate each feature. These activating examples are often cohesive and correspond to human-interpretable concepts (Cunningham et al., 2023; Templeton et al., 2024), e.g., "harmful requests". However, only examining a feature's activating examples tells us what a feature does but not what it fails to do. We might hope that a harmful request feature activates on all harmful requests, but we cannot determine this by just examining activating text. Additionally, we need to evaluate *feature sensitivity*: the probability that a feature activates on texts similar to its activating examples.

Ideally, features would have high sensitivity—consistently activating on all relevant inputs rather than arbitrary subsets. Understanding a feature's sensitivity is crucial for scoping what we can learn from the feature. If a harmful request feature has high sensitivity and activates on all harmful requests, understanding its role can reveal how the model generally processes any harmful input. If, instead, the harmful request feature has poor sensitivity, we are mainly gaining narrower insights into how the model handles the specific input that activates the feature.

In this work, we use a generation-based approach to evaluate feature sensitivity at scale. As illustrated in Figure 1, we use language models to generate text with the same semantic properties as a feature's activating examples. We then test whether the feature activates on these generated texts.

Figure 1: **Sensitivity evaluation methodology.** We extract top activating texts for each SAE feature, use GPT-4.1 to generate similar texts based on these examples, and measure how often the feature activates on the generated texts. Features with high sensitivity reliably activate on semantically similar inputs.

Our generation-based approach is more scalable and efficient than previous dataset filtering methods (Templeton et al., 2024; Turner et al., 2024). Additionally, our method avoids the need to first generate a description of the feature's activating text, removing a potential source of error compared to common automated interpretability evaluations (Paulo et al., 2024; Karvonen et al., 2025).

Our main contributions are:

- **We develop an explanation-free, scalable automated evaluation for SAE feature sensitivity**, allowing efficient evaluation of thousands of SAE features.
- **We demonstrate that sensitivity measures a new facet of feature quality** by examining its relationship to standard SAE feature metrics. Notably, we find that many interpretable features have poor sensitivity.
- **We validate our method through automated and human evaluations**, finding that when a feature fails to activate on generated text, that text genuinely resembles activating text examples according to human assessment.
- **We identify declining feature sensitivity as an additional challenge for SAE scaling**. We find that wider SAEs have lower average feature sensitivity in large-scale SAEs (up to 1M features) and across 7 different SAE variants.

## 2 RELATED WORK

### 2.1 PRIOR INVESTIGATIONS OF FEATURE SENSITIVITY

Investigating feature sensitivity requires obtaining candidate input text and checking for feature activation. Most prior work approaches this by first generating natural language explanations for features, then using those explanations to identify candidate inputs. This includes using explanations to generate new text (Huang et al., 2023; Juang et al., 2024) or to filter through existing datasets for relevant passages (Templeton et al., 2024; Turner et al., 2024).

Alternative approaches avoid natural language explanations entirely. Gao et al. (2024) fit n-grams with wildcards to activating text examples to filter datasets for test inputs. Other work evaluates whether features or groups of features can serve as high-sensitivity classifiers for a set of predefined concepts (Karvonen et al., 2024; Makelov et al., 2024; Chanin et al., 2024). Chanin et al. (2024) study feature absorption, a special instance of poor feature sensitivity with a clear cause: when features form hierarchies, sparsity incentivizes parent features (e.g., "math") to fail to activate on inputs when a more specific child feature (e.g., "algebra") activates instead.

All these approaches evaluate sensitivity with respect to some intermediate description—whether explanations, n-grams, or concept lists. Our approach evaluates sensitivity without needing to first generate such descriptions.

## 2.2 SAE Evaluation

Earlier work primarily evaluated SAEs by their reconstruction error and the interpretability of individual features (Bricken et al., 2023; Templeton et al., 2024).

Although increasing SAE width improves both reconstruction quality and feature interpretability (Karvonen et al., 2025), a growing body of research investigates problems that arise when scaling SAEs, including feature splitting (Bricken et al., 2023), feature absorption (Chanin et al., 2024), and feature composition (Leask et al., 2025). These results highlight that only optimizing for sparsity and reconstruction may not yield natural features.

Another line of work evaluates SAE latents by their utility for downstream tasks: sparse probing (Gao et al., 2024), spurious correlation removal (Marks et al., 2025a), disentangling model representations (Huang et al., 2024), and unlearning (Farrell et al., 2024). Karvonen et al. (2025) introduce SAEBench, a benchmark that aggregates many of these evaluation approaches, along with standard automated interpretability and reconstruction metrics.

## 2.3 Automated Interpretability

The standard auto-interpretability pipeline involves collecting activating text examples for a feature, prompting an LLM to generate natural language descriptions from these examples, and validating these descriptions by testing whether they enable another LLM to predict activations on new text. Bills et al. (2023) first proposed this approach for neurons, and it has since become standard for both neuron explanations (Choi et al., 2024) and SAE explanations (Paulo et al., 2024; Templeton et al., 2024; Karvonen et al., 2025).

A complementary approach evaluates explanation quality by testing whether explanations can generate new activating inputs. This approach has been used to evaluate both neuron explanations (Huang et al., 2023) and SAE feature explanations (Juang et al., 2024). Other work uses input generation to help interpretability agents test hypotheses about component activation (Shaham et al., 2025). Similar generation-based evaluation approaches have been applied beyond language models to explanations of vision neurons and other components (Singh et al., 2023; Kopf et al., 2024).

# 3 Evaluating Feature Sensitivity

## 3.1 Evaluating Feature Sensitivity Independent of Explanation

Previous work on sensitivity typically relies on some (typically natural language) description to identify test inputs (Turner et al., 2024; Juang et al., 2024). Such methods evaluate sensitivity as a function of both the model component and the corresponding explanation. When studying neurons, which are a part of the model itself, such approaches cleanly evaluate how well an explanation describes a neuron's activating inputs. However, SAE features present a more complex challenge.

Unlike neurons, SAE features are learned approximations of a model rather than intrinsic model components. Much prior work has identified and addressed limitations in feature quality arising from SAE training (Chanin et al., 2024; Leask et al., 2025; Marks et al., 2024; Bussmann et al., 2025). Because SAE features and generated feature descriptions are imperfect, evaluating feature sensitivity with explanations may struggle to distinguish between an inaccurate description of a feature and a feature failing to activate on relevant inputs.

We avoid this ambiguity by evaluating feature sensitivity without generating an explanation. As shown in Figure 1, we prompt language models with a feature's activating text examples to generate similar text samples, then measure how often the feature activates on these new texts. For a feature to achieve high sensitivity, it must consistently activate on novel inputs that human judges find indistinguishable from the original activating examples. This approach effectively measures sensitivity as if we had a perfect explanation—one precise enough to generate indistinguishable examples but nothing broader.

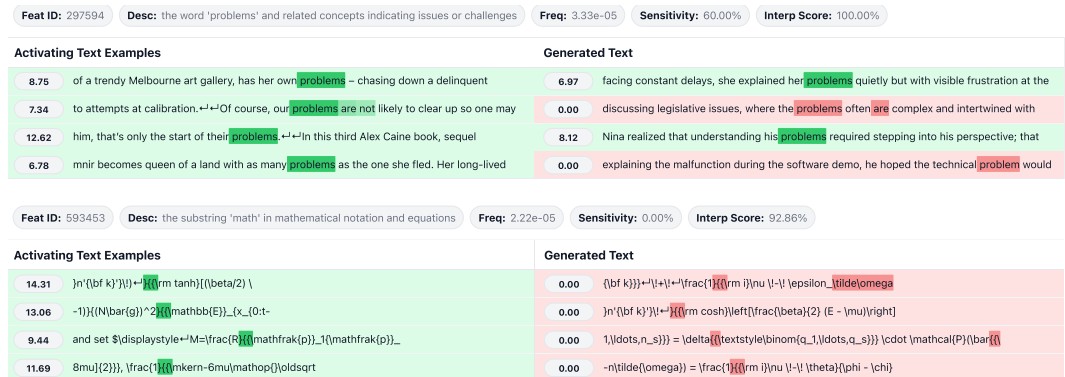

Figure 2: **Interpretable features with moderate and low sensitivity.** Feature activations are shown on top activating texts (left) and on LLM-generated texts from our evaluation (right). Generated text is formatted to indicate tokens expected to activate the feature. These are highlighted when the feature remains inactive.

## 3.2 METHOD DETAILS

Our sensitivity evaluation approach consists of four steps: (1) collect activating text examples for each feature, (2) generate new texts similar to these examples using an LLM, (3) evaluate if the feature is active on these new texts, and (4) compute sensitivity score as the fraction of new generated texts which successfully cause the feature to activate. In the paragraphs below, we provide additional details for the first two steps. Figure 2 shows examples of text generated by our evaluation.

**Collecting Activating Text:** We sample 2 million tokens of candidate texts from large text corpora. The corpus is OpenWebText (Gokaslan et al., 2019) for SAEBench evaluations and the Pile-uncopyrighted subset (Gao et al., 2020) for GemmaScope evaluations. We evaluate feature activation on sequences of 128 tokens, following the example collection methodology used in (Karvonen et al., 2025). When a feature activates, we extract the activating example by including 10 tokens preceding and 10 tokens following the activating token. For each feature, we collect 15 activating text examples: 10 top activating examples and 5 importance-weighted samples by activation magnitude.

**Generating New Texts:** We provide activating text examples when prompting an LLM. We do *not* use any natural language descriptions of the feature in the prompt. In preliminary experiments, adding automated feature descriptions reduced the probability that generated text would activate the feature. From inspecting samples, we believe this is due to automated descriptions that are sometimes overly general and imprecise. For each feature, we use a single query to generate 10 new text samples. We found that a single query produced more diverse outputs than multiple independent queries. The full prompts are included in Appendix A. We use GPT-4.1-mini (OpenAI et al., 2024) for the generation step. We found that it produced text comparable to GPT-4.1, while GPT-4.1-nano struggled to complete the generation task.

**Method Assumptions:** Our method relies on several key assumptions. First, we require that our collected examples adequately capture each feature's behavior, which we ensure by following standard approaches for collecting activating examples and filtering out features that fail to activate on truncated text. Details of filtering are described in Section 3.3. Second, we assume that generated texts share whatever semantic property triggers feature activation, which we validate through human evaluation in Section 5.1. Third, we assume generated samples are sufficiently novel and diverse to serve as valid tests of sensitivity, which we verify in Section 5.2.

## 3.3 FILTERING SAE FEATURES

We limit our study to SAE features that meet two criteria. First, we only evaluate features for which we can collect at least 15 activating text samples from 2 million tokens, which filters out rare features. Second, we found that many features fail to activate on their own truncated examples, so we filter for features where at least 90% of the shortened text snippets still activate the feature. This

filtering may bias our analysis toward simpler features, but it ensures that features failing to activate on generated text genuinely reflect poor sensitivity, rather than an artifact of sample text truncation. The fraction of filtered features increases substantially with SAE width. For smaller SAEBench SAEs (width 4k to 65k), we exclude 35% of features on average. For GemmaScope SAEs, this ranges from 51% for 65K width SAEs to 79% for 1M width SAEs. Detailed filtering statistics and results with different cutoffs are shown in Appendix B.

## 4 FEATURE SENSITIVITY CAPTURES NOVEL ASPECTS OF FEATURE BEHAVIOR

We begin by examining the relationship between our feature sensitivity metric and standard SAE feature evaluation metrics. For this, we study the canonical (width 1M, sparsity 107) GemmaScope (Lieberum et al., 2024) SAE for the layer 12 residual stream of Gemma 2 2B (Team et al., 2024). We sampled 10,000 SAE features. After filtering per Section 3.3, 2,061 remained for analysis.

We show the distribution of sensitivities across all features in Figure 3a. Most features score well on sensitivity, but the features span all sensitivity scores, showing meaningful variation in feature quality when measured via sensitivity.

Next, we examine three key feature properties for comparison. First, we look at feature interpretability, which we measure using the automated interpretability evaluation of (Karvonen et al., 2025). Second, we examine feature frequency, which is how often features have nonzero activation. Third, we compute the maximum decoder cosine similarity between a feature's decoder vector and all other feature decoder vectors. High similarity may reflect undesirable feature composition or entanglement (Bussmann et al., 2025).

The three scatter plots of feature sensitivity and each property (Figures 3b, 3c, and 3d) confirm that feature sensitivity is distinct from existing other metrics. We find weak correlations of sensitivity with frequency ($\rho = -0.06$) and decoder cosine similarity ($\rho = 0.06$), and a stronger correlation between sensitivity and interpretability score ($\rho = 0.24$). The overall weak correlations with existing metrics are encouraging—they suggest that sensitivity captures a novel and complementary dimension of feature quality rather than simply replicating existing evaluations.

Although feature interpretability and feature sensitivity are correlated, they often disagree. When examining features with high sensitivity but low auto-interpretability scores, we find this mainly reflects noise in the automated evaluation—these features appear qualitatively interpretable upon inspection. More importantly, *we find many interpretable features exhibit poor sensitivity*. Among 1347 features with auto-interpretability scores $\geq 0.9$, 82 have sensitivity $\leq 0.5$, and 23 have sensitivity $\leq 0.2$. Figure 2 shows examples of interpretable features with moderate and low sensitivity, with additional examples in Appendix G. Spot checking these features shows that our evaluation-generated text resembles activating text but fails to activate the feature, suggesting that our method has indeed found interpretable features that have poor sensitivity. In the next section, we validate this rigorously via human evaluation.

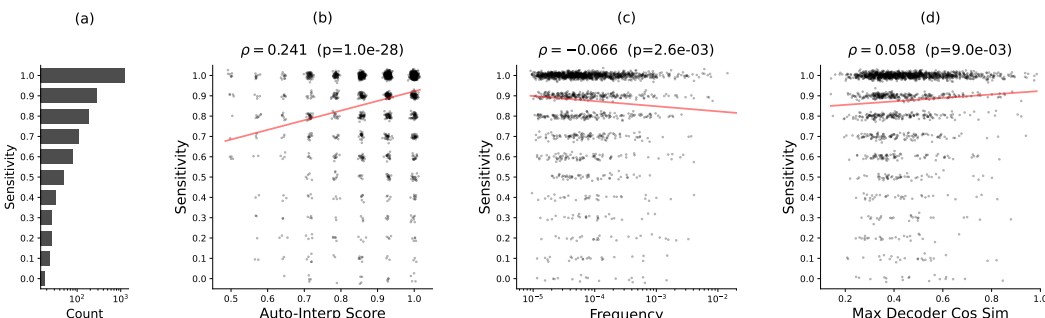

Figure 3: **GemmaScope SAE feature sensitivity distributions.** The distribution of feature sensitivity and scatter plots showing joint distributions of sensitivity with auto-interpretability, frequency, and maximum decoder cosine similarity. Sensitivity scores in scatter plots are plotted with y-jitter for visualization. Correlation coefficients and p-values are shown at the top of each scatter plot.

# 5 VERIFYING THE AUTOMATED SENSITIVITY EVALUATION

We validate that our automated sensitivity evaluation is reliable through two analyses: (1) human evaluation of sample similarity and (2) automated evaluations of sample novelty and diversity.

## 5.1 BLINDED HUMAN EVALUATION

The goal of the human evaluation is to check if human annotators agree that the LLM generations are indeed consistent with the feature concept, and therefore appropriate for scoring feature sensitivity.

Human annotators judged 102 examples in total. Each example consists of several activating text examples for a feature along with one new text sample. The new text can be one of three categories: another activating text example for the feature (20%, positive control), a generated text for a random other feature (20%, negative control), or a text generated by our method that failed to activate the feature (60%). The category is not revealed to the human annotator. The human annotator is then asked to classify whether the new text is "indistinguishable", "closely related", "weakly related", or "unrelated" to the provided activating text examples. A sample dashboard for the human evaluation is shown in Figure 4b. We only include features with high auto-interpretability ($\geq 0.9$). This allows the study to focus on verifying cases where we might be most skeptical of low sensitivity results a priori. Additionally, interpretable features are easier for human annotators to assess.

Results are shown in Figure 4a. *Generated text achieves relevance ratings nearly matching ground truth, confirming that low sensitivity evaluations reflect poor sensitivity rather than poor generation.* Human annotators rate our method's generated texts ($n = 62$) nearly as relevant to the feature as the ground truth texts: 79% of generated texts are rated "indistinguishable", compared to 83% of ground truth activating texts. Only one out of 62 generated texts is rated "unrelated". Additionally, annotators correctly scored controls: positive control texts ($n = 24$) are rated "indistinguishable" or "closely related" 96% of the time, while all negative control texts ($n = 16$) are rated "unrelated" or "weakly related".

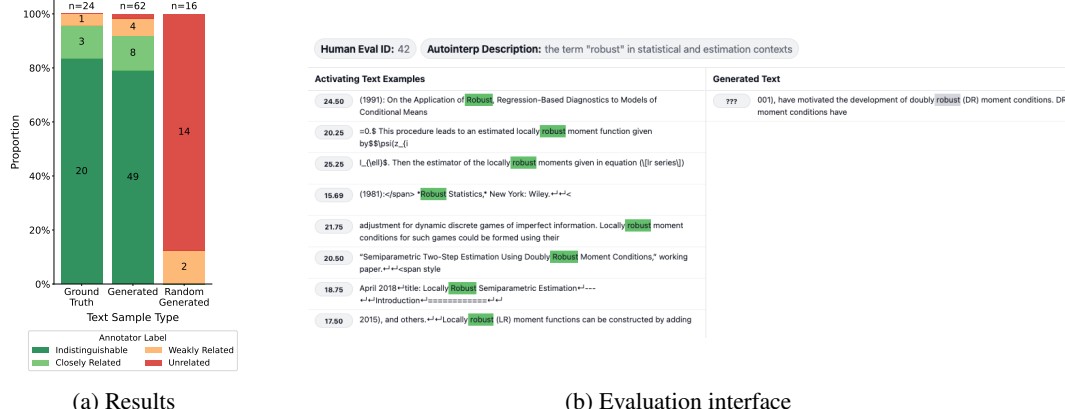

(a) Results        (b) Evaluation interface

Figure 4: **Human evaluation validates our method.** (a) Human evaluation of 102 text samples across three conditions: true activating text examples (positive control), text generated for random features (negative control), and text generated by our evaluation that failed to activate features. (b) The interface shows feature activating examples alongside generated text for evaluation, with annotators rating similarity.

## 5.2 SAMPLE NOVELTY AND DIVERSITY

The goal of this analysis is to check that (1) our generated texts were not copying the activating examples, i.e., the diversity between each generated text and the top-activating texts is sufficiently high, and (2) our generated texts covered a wide range of feature expression, proxied by checking that the diversity between generated texts is sufficiently high.

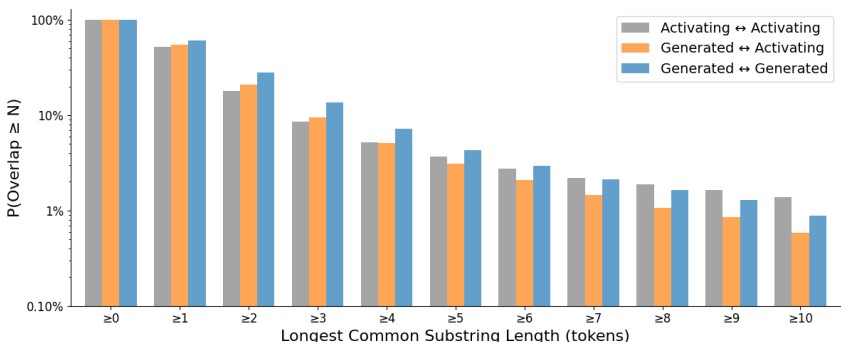

Figure 5: **Text diversity validation.** Probability that the longest common substring length is $\geq$ N tokens. We compare: two activating text examples for the same feature (gray), one generated text and one activating text example for the same feature (orange), and two generated text samples for the same feature (blue).

We assess text diversity by measuring the longest common substring length across three comparisons: (1) between generated text with activating examples to evaluate copying, (2) between pairs of activating examples to establish baseline overlap levels, and (3) between pairs of generated texts to assess diversity within our generations. Also note that we checked for longest substring match ending on the activating tokens, since only tokens before the activating part contribute to the activation.

Figure 5 shows the complementary cumulative distribution function (CCDF) for longest common substring lengths. Each bar shows the fraction of text pairs with overlap $\geq N$ tokens: gray bars show overlap between activating examples (baseline), orange bars show overlap between generated and activating texts (testing for copying), and blue bars show overlap between generated texts (testing for diversity).

The first reassuring observation is that a generated text and an activating text example are less likely to have a long overlap than two activating examples (3.1% v.s. 3.7% at $\geq$ 5 tokens). On the other hand, a generated text and an activating text example are more likely to contain a short overlap than two activating examples (20.8% v.s. 18.0% at $\geq$ 2 tokens). This indicates that our generated texts occasionally use short verbatim sequences from the examples but avoid copying long passages.

Two generated texts are slightly more likely to have overlap than the baseline between activating examples, with 27.9% probability of $\geq$ 2 token overlap and 4.3% at $\geq$ 5 tokens. This reveals that pairs of generations show somewhat lower diversity, though the difference is modest. This overlap pattern likely reflects LLM preferences for common word choices and short phrases rather than wholesale copying. While generation diversity can be improved, there are no pathological issues with extended substring duplication.

## 6 EVALUATING FEATURE SENSITIVITY ACROSS SAEs

Having explored the sensitivity of features within a single SAE and having confirmed that our evaluation method is reliable, we now turn to evaluating the average feature sensitivity across different SAE sizes and architectures.

### 6.1 RESULTS ON LARGE GEMMASCOPE SAEs

The GemmaScope suite of twenty nine JumpReLU SAEs range in size from 65K to 1M features and range in sparsity from 20 to 200 (Lieberum et al., 2024). These SAEs are trained to reconstruct the layer 12 residual stream of Gemma 2-2B (Team et al., 2024). For each SAE in GemmaScope, we collect activating texts for 2500 features, then apply the filtering criteria described in Section 3.2 and Appendix B before computing sensitivity.

Figure 6 shows the effect of dictionary width and sparsity on feature sensitivity. At a fixed dictionary size, sensitivity increases as sparsity increases. Strikingly, *as SAE width increases, average feature*

378
379
380
381
382
383
384
385
386
387
388
389

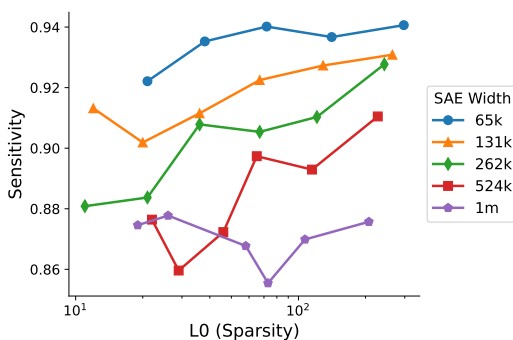

390
391
392

Figure 6: **Average Feature Sensitivity of GemmaScope SAEs.** For each dictionary size, we plot the feature sensitivity of SAEs trained at that size at different sparsities. Wider SAEs have worse average feature sensitivity. We also see that feature sensitivity is slightly increasing with sparsity.

393
394
395
396
397
398

*sensitivity decreases*. Concretely, 65K width SAEs have average feature sensitivities ranging from 0.92 to 0.94, while 1M width SAEs have feature sensitivities ranging from 0.85 to 0.87. Additionally we find that at a fixed width, SAEs with high L0 - more active features - have higher average feature sensitivity. In Appendix E we show that these two trends hold after controlling for feature frequency.

399
400

## 6.2 RESULTS ON DIVERSE SAE ARCHITECTURES

401
402
403
404
405
406
407

Having found these scaling trends on GemmaScope JumpReLU models, we next test whether they generalize across different model families and SAE architectures. We evaluate SAEs from the SAEBench collection (Karvonen et al., 2025), which includes 7 different SAE architectures trained on both Pythia-160M (Biderman et al., 2023) and Gemma-2-2B (Team et al., 2024) models. While these SAEs are much smaller in scale than GemmaScope, they allow us to validate our findings across SAE variants and model architectures. For each SAE studied here, we collect activating text for 1000 features, then filter as before.

408
409
410
411
412
413

We show the relationship between sparsity and sensitivity on the largest SAEs in this suite (65k width) in Figure 7. While the results are noisier due to smaller sample sizes, we see a general trend of sensitivity increasing with sparsity across model and SAE variants. While noise prevents us from making strong claims about sensitivity differences between each of the SAE architectures, vanilla ReLU SAEs consistently show low sensitivity, performing worst on Gemma-2-2B and among the worst variants on Pythia-160M.

414
415
416
417

Next, we examine how dictionary size affects sensitivity across architectures. To control for sparsity, we select SAEs with L0 closest to 80 (exactly 80 for top-K SAEs, closest available for other variants). The results in Figure 8 confirm that wider SAEs consistently show worse sensitivity across

418
419
420
421
422
423
424
425
426
427
428

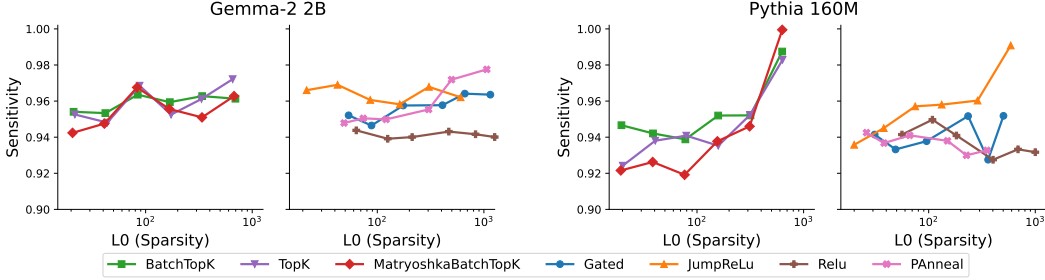

429
430
431

Figure 7: **Average Sensitivity vs. Sparsity for Gemma-2-2b and Pythia-160m SAEs** This plot shows the average sensitivity of different Sparse Autoencoder (SAE) types plotted against their sparsity. We use the widest 65k width SAEs for all architectures. Each line represents a different SAE architecture.

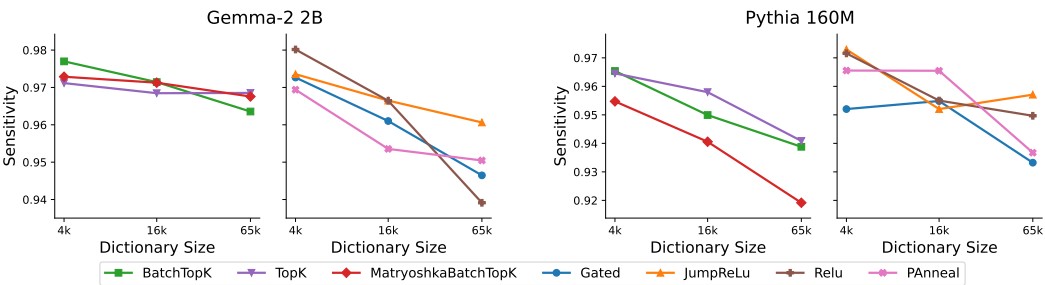

Figure 8: **Average Sensitivity vs. Dictionary Size for Gemma-2-2b and Pythia-160m SAEs** This plot shows the average sensitivity of different Sparse Autoencoder (SAE) types plotted against their dictionary size. We select SAEs with L0 closest to 80 (exactly 80 for top-K SAEs, closest available for other variants). Each line represents a different SAE architecture.

all tested architectures. Notably, *Matryoshka SAEs also exhibit negative scaling with sensitivity*, despite being specifically designed to address scaling challenges in SAEs (Bussmann et al., 2025).

## 7  DISCUSSION AND CONCLUSION

We developed a scalable pipeline that generates texts similar to SAE feature activating examples. We validate through human evaluation that these generated texts are genuinely similar—humans judge them as indistinguishable from actual activating examples. We use this pipeline to evaluate individual features and average sensitivity of features in an SAE. At the feature level, we found that many interpretable features have poor sensitivity, broadening our notion of what makes a high-quality SAE feature. At the SAE level, we found that average feature sensitivity consistently decreases as SAE width increases, identifying a new challenge for scaling SAEs. Taken together, our work helps develop feature sensitivity as a new axis to evaluate both individual features and SAE variants.

### 7.1  LIMITATIONS AND FUTURE WORK

Beyond evaluation, our pipeline opens new directions for exploratory analysis. Studying feature activations on text generated by our pipeline could enable more fine-grained studies of the boundaries separating activating from non-activating inputs for a given feature. This approach could also enable the study of groups of features that may collectively represent specific concepts with high sensitivity. Additionally, our pipeline and sensitivity evaluation can be applied to any model component that activates on input text. Future research could examine sensitivity in thresholded neurons, transcoders (Dunefsky et al., 2024), and cross-layer transcoders (Ameisen et al., 2025).

Our evaluation was limited to frequently occurring features (15+ times in 2M tokens), which biases our analysis toward common features and misses potentially important rare features. We filter for features that remain active when truncated activating text is used, potentially biasing toward simpler features that don't depend on longer contexts. Future work can directly scale up this evaluation by studying less frequent features and using longer text snippets. Additionally, we don't meaningfully incorporate information about the magnitude of feature activation in each passage. We would be excited by future work that incorporates activation strength into studies of SAE features, either in the context of sensitivity or broader evaluation.

## REFERENCES AND REPRODUCIBILITY

We have uploaded our code anonymously as supplementary material for the review process at `https://anonymous.4open.science/r/sae-sensitivity-8247`. For the camera-ready version, we will release it publicly on GitHub. We use publicly available SAEs from SAEBench (Karvonen et al., 2025) and GemmaScope (Lieberum et al., 2024), and publicly available data from OpenWebText (Gokaslan et al., 2019) and the Pile (Gao et al., 2020). Implementation details are provided in Section 3.2, with evaluation prompts in Appendix A.

## ETHICS STATEMENT

Our work evaluates interpretability methods and does not directly enable new capabilities or applications. We recognize that improved understanding of neural networks could have dual-use implications, potentially aiding both safety research and capabilities development. Our human evaluation was conducted by the authors themselves, avoiding concerns regarding external participants. All experiments used publicly available models and datasets.

## USE OF LARGE LANGUAGE MODELS

We used Claude Code and ChatGPT as general-purpose tools to assist with implementing experiment code, generating plots, formatting the paper, and revising text. All LLM outputs were carefully verified and checked by the authors. The research idea, experimental design, and conclusions were developed by the authors without LLM assistance. The authors take full responsibility for all content in this work.

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

# A    EVALUATION PROMPTS

---

**System**

You are a meticulous AI researcher conducting an important investigation into a specific feature inside a language model that activates in response to text inputs. Your overall task is to generate additional text samples that cause the feature to strongly activate.

You will receive a list of text examples on which the feature activates. Specific tokens causing activation will appear between delimiters like {{this}}. Consecutive activating tokens will also be accordingly delimited {{just like this }}. If no tokens are highlighted with {}, then the feature does not activate on any tokens in the input.

Note: features activate on a word-by-word basis. Also, feature activations can only depend on words before the word it activates on.

---

**User**

Consider the feature that activates when the given examples below are present. Your task is to generate text samples that strongly activate this feature. Study the examples carefully to identify both their shared and varying traits. Your generated samples should:
- Preserve any consistent traits, patterns, or constraints present across all examples
- Match the diversity level shown in the examples---neither more diverse nor more uniform
- Vary along the same dimensions that the examples vary (e.g., if examples differ in tone but share a topic, maintain that pattern)
- Avoid introducing new types of variation not present in the example set
- Avoid collapsing into repetitive or overly similar outputs

Generate exactly 11 new samples separated by <SAMPLE_SEPARATOR/>. Note that the feature may involve semantic content, grammatical structures, abstract concepts, specific named entities (e.g., people, organizations, locations), or formatting elements like newlines, punctuation, citations, or special characters, for example, {{\n}}, or {{←↩}} represent newlines, {{,}} represents commas, {{-}} represents hyphens, etc that are activating the feature. Present each sample without numbering or bullets.
Important: place <SAMPLE_SEPARATOR/> between generated samples.

See the following 15 examples that activate the feature, separated by <SAMPLE_SEPARATOR/>:

<SAMPLE_SEPARATOR/>
count();

static const char*{{ resource}}{{_}}to_cstring(const char*{{ resource}}

<SAMPLE_SEPARATOR/>

What is the Java equivalent of JavaScript's{{ resource}} folder?↩→↩My Wicket web application contains

<SAMPLE_SEPARATOR/>
side-effect: since the check isn't so{{ resource}}↩intensive, you can set the time between checks
...

---

```
Assistant

"void free{{ resource}}Memory(void* ptr);
static const char* load_{{ resource}}Path(const char*{{ resource}});"

<SAMPLE_SEPARATOR/>

"How to configure the {{ resource}} directory in a Python Flask application?
I'm trying to serve static files from the"

<SAMPLE_SEPARATOR/>

warning: avoid heavy computation in the{{ resource}} allocation phase, it may
slow down startup.

<SAMPLE_SEPARATOR/>
...
```

## B   FEATURE FILTERING DETAILS

We evaluated 112 SAEs from the SAEBench dataset and 29 from the GemmaScope dataset. The SAEBench set spans seven SAE families—BatchTopK, MatriyoshkaTopK, TopK, JumpReLU, ReLU, Gated, and PAnneal—whereas all GemmaScope SAEs are JumpReLU.

During the study, we observed that some activation texts distributed with SAEBench do not consistently activate their associated SAE features, likely due to truncation. To address this, we computed the *activation rate in truncated example text* for each feature, defined as the proportion of published activation texts that reliably elicit the feature. Features with an activation rate below 90% were excluded from our analysis. Table 1 and Table 2 reports the impact of this filtering on our study.

In Figure 9 we show our main Gemmascope results with different filtering thresholds. We see that for all choices of threshold, our main results hold.

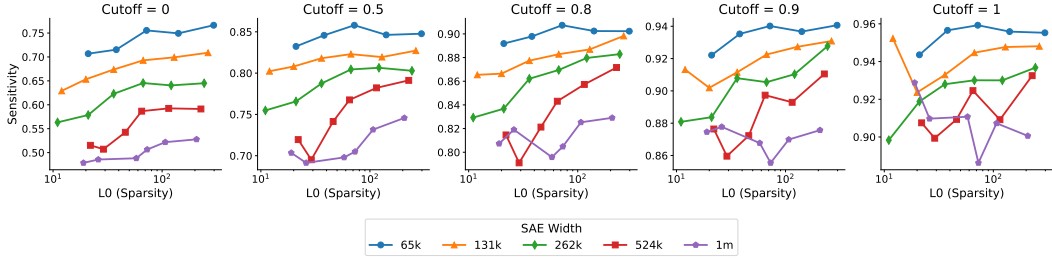

Figure 9: **Robustness to Feature Selection Cutoffs.** GemmaScope scaling results shown with different shortened text activation filter cutoffs. Our main results are robust to the choice of cutoff threshold, demonstrating that the observed scaling trends are not artifacts of our feature selection criteria.

| Model | SAE Type | No. SAEs | Avg. No. Feat. | Avg. Sens. | 90% Activation Rate Threshold | | | |
|---|---|---|---|---|---|---|---|---|
| | | | | | Avg. No. Remain. | Avg. Sens. | % Feat. Excluded | % Sens. Change |
| Gemma-2-2B | All, 16k | 7 | 998 | 0.875 | 704 | 0.982 | 29.4% | 12.2% |
| | All, 4k | 7 | 999 | 0.918 | 801 | 0.984 | 19.8% | 7.2% |
| | BatchTopK, 65k | 6 | 969 | 0.814 | 562 | 0.980 | 42.1% | 20.6% |
| | Gated, 65k | 6 | 981 | 0.826 | 562 | 0.978 | 42.7% | 18.4% |
| | JumpReLu, 65k | 6 | 981 | 0.834 | 597 | 0.979 | 39.2% | 17.4% |
| | MatryoshkaBatchTopK, 65k | 6 | 964 | 0.776 | 485 | 0.979 | 49.7% | 26.3% |
| | PAnneal, 65k | 6 | 997 | 0.893 | 749 | 0.986 | 24.9% | 10.7% |
| | Relu, 65k | 6 | 994 | 0.848 | 646 | 0.983 | 35.0% | 16.0% |
| | TopK, 65k | 6 | 972 | 0.820 | 574 | 0.979 | 41.0% | 19.7% |
| Pythia-160M | All, 16k | 7 | 995 | 0.569 | 417 | 0.981 | 58.1% | 137.8% |
| | All, 4k | 7 | 1299 | 0.908 | 1029 | 0.986 | 21.0% | 8.6% |
| | BatchTopK, 65k | 6 | 978 | 0.850 | 674 | 0.987 | 30.9% | 16.1% |
| | Gated, 65k | 6 | 994 | 0.786 | 522 | 0.978 | 47.5% | 24.6% |
| | JumpReLu, 65k | 6 | 995 | 0.869 | 727 | 0.985 | 26.9% | 13.4% |
| | MatryoshkaBatchTopK, 65k | 6 | 968 | 0.817 | 624 | 0.986 | 35.1% | 20.8% |
| | PAnneal, 65k | 6 | 998 | 0.838 | 627 | 0.985 | 37.2% | 17.6% |
| | Relu, 65k | 6 | 997 | 0.834 | 633 | 0.984 | 36.5% | 18.1% |
| | TopK, 65k | 6 | 978 | 0.845 | 667 | 0.987 | 31.7% | 17.3% |
| **ALL** | | **112** | **1006** | **0.830** | **648** | **0.983** | **35.6%** | **18.4%** |

Table 1: SAE filtering statistics showing the impact of excluding features with activation rate below 90% in truncated example text. Columns show the model, SAE type, number of SAEs, average features per SAE before filtering, average sensitivity before filtering, and the effects after applying the 90% threshold: remaining features, new sensitivity, percentage excluded, and percentage sensitivity change.

| Width | No. SAEs | Features Evaluated | Features Remaining | % Excluded |
|---|---|---|---|---|
| 65k | 5 | 2336 | 1144 | 51.0% |
| 131k | 6 | 2438 | 990 | 59.4% |
| 262k | 6 | 2381 | 798 | 66.6% |
| 524k | 6 | 2339 | 629 | 73.2% |
| 1M | 6 | 2278 | 485 | 78.8% |

Table 2: GemmaScope filtering statistics with 90% activation rate cutoff. All SAEs are JumpReLU trained on Gemma-2-2B layer 12 residual stream. Wider SAEs show increased feature exclusion rates.

## C NORMALIZED SENSITIVITY METRICS

Our main analysis filters features based on their activation rate in truncated example text, excluding 35–79% of features with exclusion rates increasing for wider SAEs (Table 2). To further verify that filtering does not explain our scaling results, we introduce two normalized metrics that account for varying truncated activation rates without hard filtering.

Let $p$ denote a feature's activation rate on truncated example text and $q$ denote its activation rate on generated text. We define the **sensitivity ratio** as $\min(q/p, 1)$ (1 if $p = 0$), measuring what fraction of a feature's truncated-text activation rate is achieved on generated text. We define the **sensitivity gap** as $\min(p - q, 0)$, measuring the drop in activation rate between truncated activating examples and generated evaluation text.

Figure 10 shows both metrics averaged across GemmaScope SAE features. These metrics are computed *without any feature filtering*. Wider SAEs show lower sensitivity ratios and larger sensitivity gaps, reproducing our main finding that sensitivity declines with SAE width.

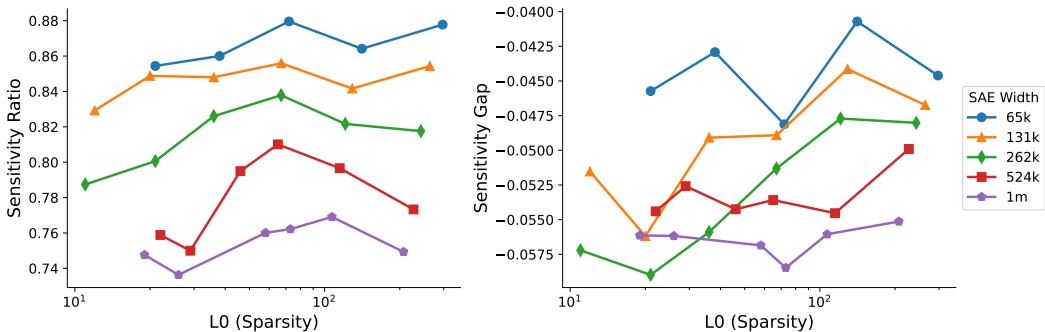

Figure 10: **Normalized sensitivity metrics across SAE widths.** Left: Sensitivity Ratio $(\min(q/p, 1))$ measures the fraction of a feature's truncated activation ceiling achieved by generated text. Right: Sensitivity Gap $(\min(p - q, 0))$ measures the absolute difference between truncated and generated activation rates. Both metrics show that wider SAEs have lower sensitivity, confirming that our main finding is not an artifact of feature filtering.

# D  SENSITIVITY AND DOWNSTREAM TASK PERFORMANCE

We conducted a preliminary analysis examining the relationship between sensitivity and downstream task performance across 42 SAEs trained on Gemma-2-2B from SAEBench (Karvonen et al., 2025). These SAEs span multiple architectures and sparsity levels.

Figure 11 shows scatter plots of average SAE sensitivity against performance on five downstream tasks. Sensitivity shows significant positive correlations with SCR ($r = 0.43$, $p = 0.005$), TPP ($r = 0.33$, $p = 0.036$), and RAVEL ($r = 0.43$, $p = 0.005$), and a significant negative correlation with Unlearning ($r = -0.37$, $p = 0.017$). The correlation with Sparse Probing was not significant ($r = 0.23$, $p = 0.14$).

We wish to emphasize that these results are exploratory and should be interpreted with caution. The SAEs vary in architecture and sparsity, and we do not control for other SAE attributes (e.g., L0, reconstruction quality) or correct for multiple comparisons. Future work could examine whether sensitivity provides unique predictive value beyond existing metrics.

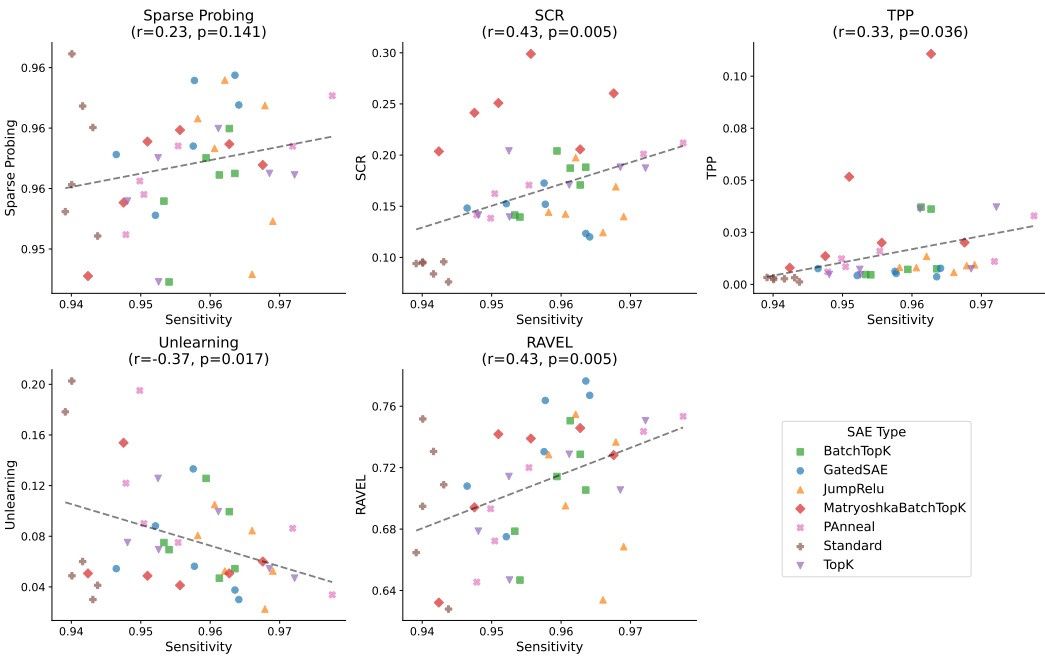

Figure 11: **Sensitivity correlates with downstream task performance.** Scatter plots of average SAE sensitivity against performance on five SAEBench tasks across 42 SAEs trained on Gemma-2-2B. Sensitivity shows significant correlations with SCR, TPP, RAVEL, and Unlearning. Results are exploratory; see text for caveats.

# E  CONTROLLING FOR FEATURE FREQUENCY

To ensure that our sensitivity results are not confounded by differences in feature frequency across SAE widths, we repeated our GemmaScope analysis with frequency-weighted sampling. Different width SAEs may have systematically different feature frequency distributions, which could potentially influence average sensitivity measurements.

## E.1  WEIGHTING METHODOLOGY

We re-weighted features so that each SAE has the same effective frequency distribution. Specifically, for each SAE, we:

1. Computed the frequency distribution of features across all SAEs in our study

2. Determined a target frequency distribution (the average distribution across all SAE widths)

3. Assigned weights to each feature inversely proportional to its frequency's representation in the SAE relative to the target distribution

4. Re-computed average sensitivity using these weights

Figure 12 illustrates this re-weighting process, showing how features at different frequencies are weighted to achieve a uniform distribution across SAEs.

## E.2  RESULTS WITH FREQUENCY CONTROL

Figure 13 shows the results after applying frequency weighting. Explicitly controlling for feature frequency via reweighting does not change our main results. Wider SAEs show lower average feature sensitivity. At a given width, SAEs with more active latents have higher sensitivity. This confirms that our main results are not an artifact of frequency distribution differences across SAE widths or sparsities.

The similarity between these frequency-controlled results and our main findings (Figure 6) demonstrates that the sensitivity-width tradeoff is a robust phenomenon independent of feature frequency distributions.

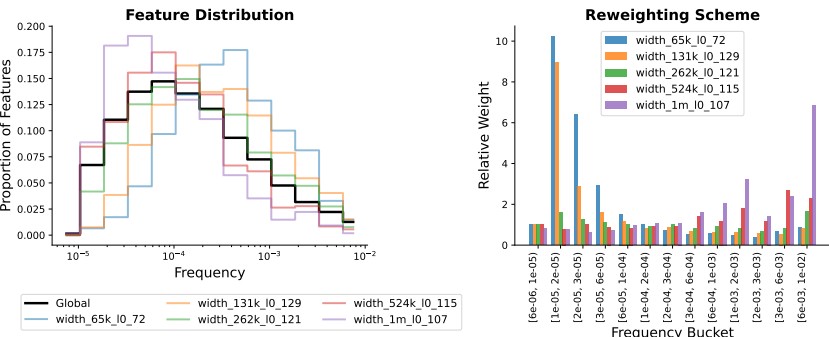

Figure 12: **Frequency re-weighting methodology.** Visualization of how features are re-weighted to control for frequency differences across SAE widths.

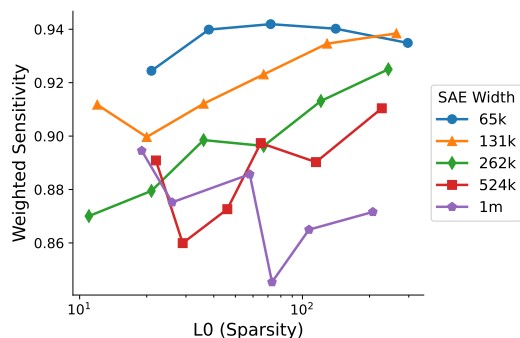

Figure 13: **Feature sensitivity with frequency weighting.** Average sensitivity across GemmaScope SAEs after re-weighting to control for feature frequency. The declining sensitivity with width persists, confirming our main findings.

## F  PRECEDING TOKEN LENGTH ANALYSIS

When we look through generated text that fails to activate the feature, we occasionally see cases where the text that intends to activate the feature appears very early in the sequence. We wanted to check if this early positioning of feature-related text was the cause of the feature failing to activate. To investigate this, we collected all generated texts and, for each one, looked for the first token that the model annotated with curly braces—this annotation indicates where the model was intending for the feature to activate, which we call target tokens. In Figure 14, we show the distribution of where the target token appears in the generated text.

We found that generated texts indeed often have relatively short prefixes leading up to the target token. For example, in 1.5% of generations, the target token is actually the first token of the genera- tion, and in around 30% of generations, the target token is preceded by 5 or fewer tokens. However, we see that even in generated text samples where the target token occurs early in the sample, most of these samples successfully activate the feature. We do note that the proportion of generated text which fails to activate the feature is higher in generations with shorter prefixes. This represents a slight limitation of our evaluation that could be improved with better prompting and instructions, though the high success rate of feature activation even with short or no prefixes suggests that the bias does not significantly compromise our evaluation.

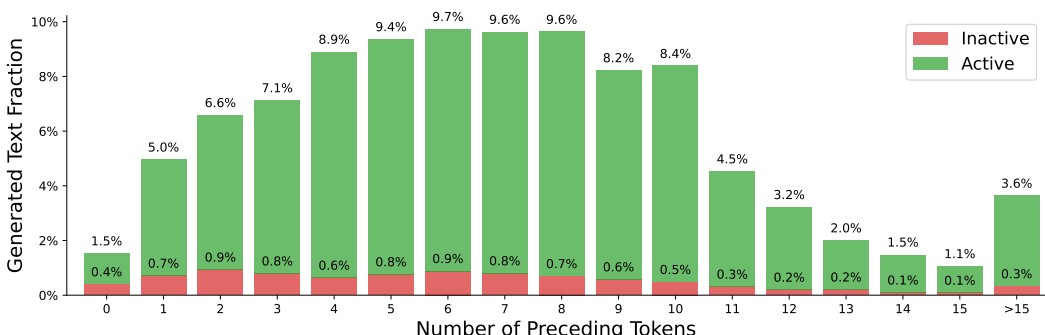

Figure 14: **Target Token Position and Feature Activation Success.** For each generated text sam- ple, we identify the target token expected to activate the feature. The chart shows the number of tokens which occur in the generated text before the target token. Bars are colored based on whether the generated text successfully activated the feature (green) or failed to activate (red).

## G  ADDITIONAL FEATURE EXAMPLES

We present additional feature dashboards showing interpretable features with zero sensitivity (Figure 15), interpretable features with moderate sensitivity (Figure 16), and features with high sensitivity but low automated interpretability scores that appear qualitatively interpretable (Figure 17). Each dashboard displays 4 out of 15 activating text examples and 4 out of 10 generated text examples.

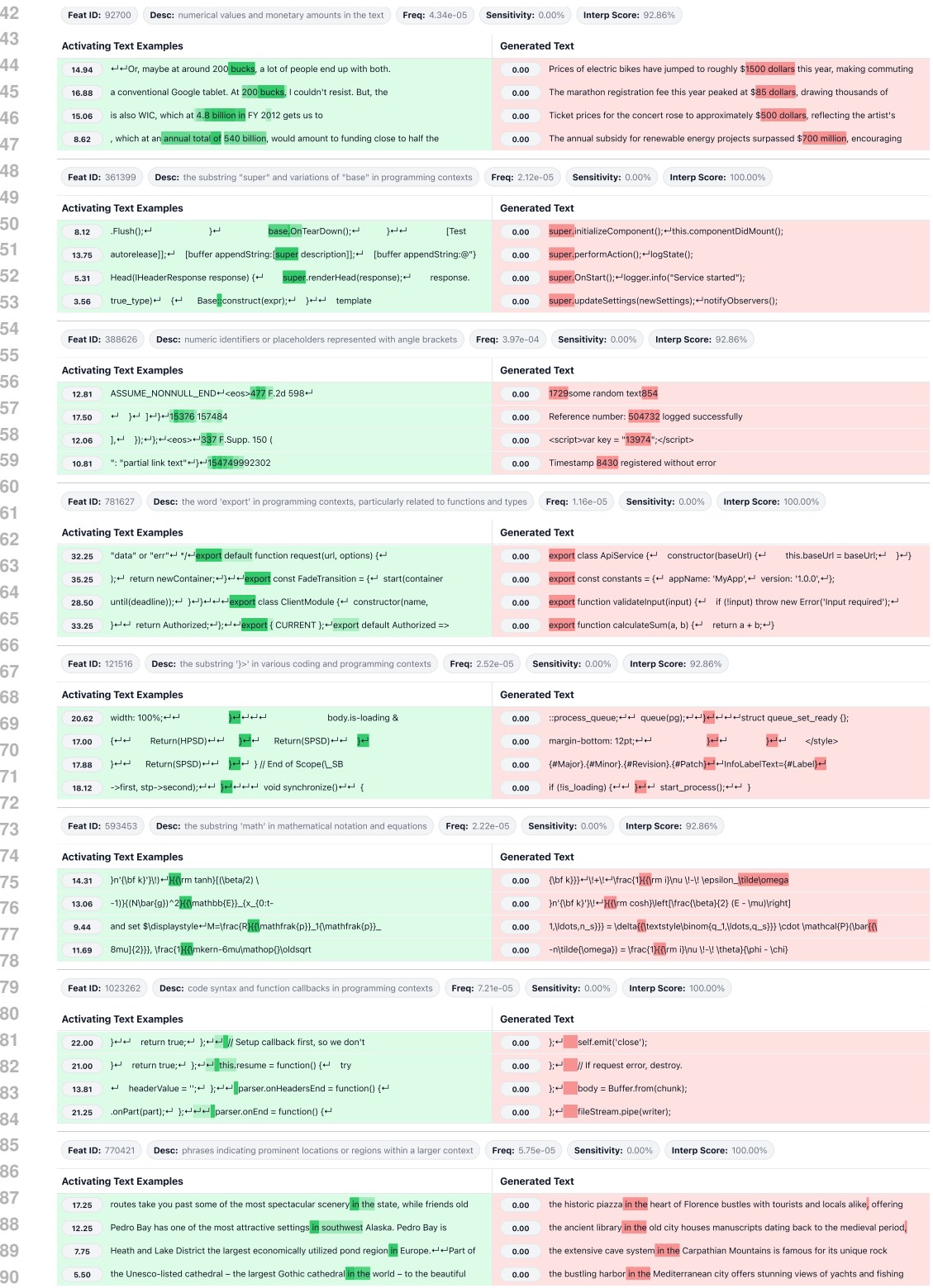

Figure 15: All 8 SAE features studied in Figure 3 that have sensitivity score 0 and auto-interp score over 0.9. For 3 of these features, low sensitivity may be due to generated passages immediately starting with the text intended to activate the feature.

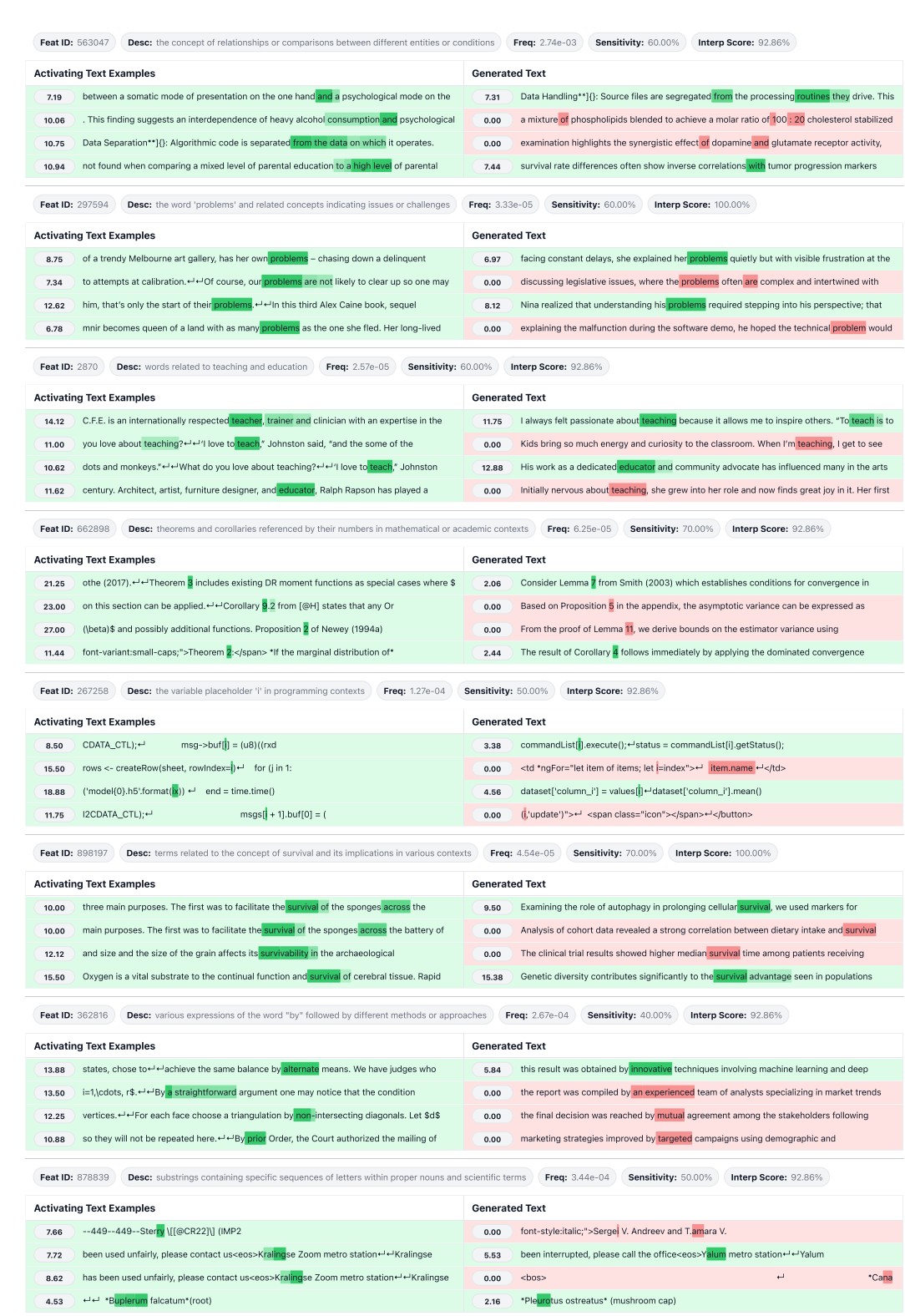

Figure 16: 8 randomly sampled features from those studied in Figure 3 that have sensitivity score between 0.4 and 0.7 and high ($\geq 0.9$) auto-interp score.

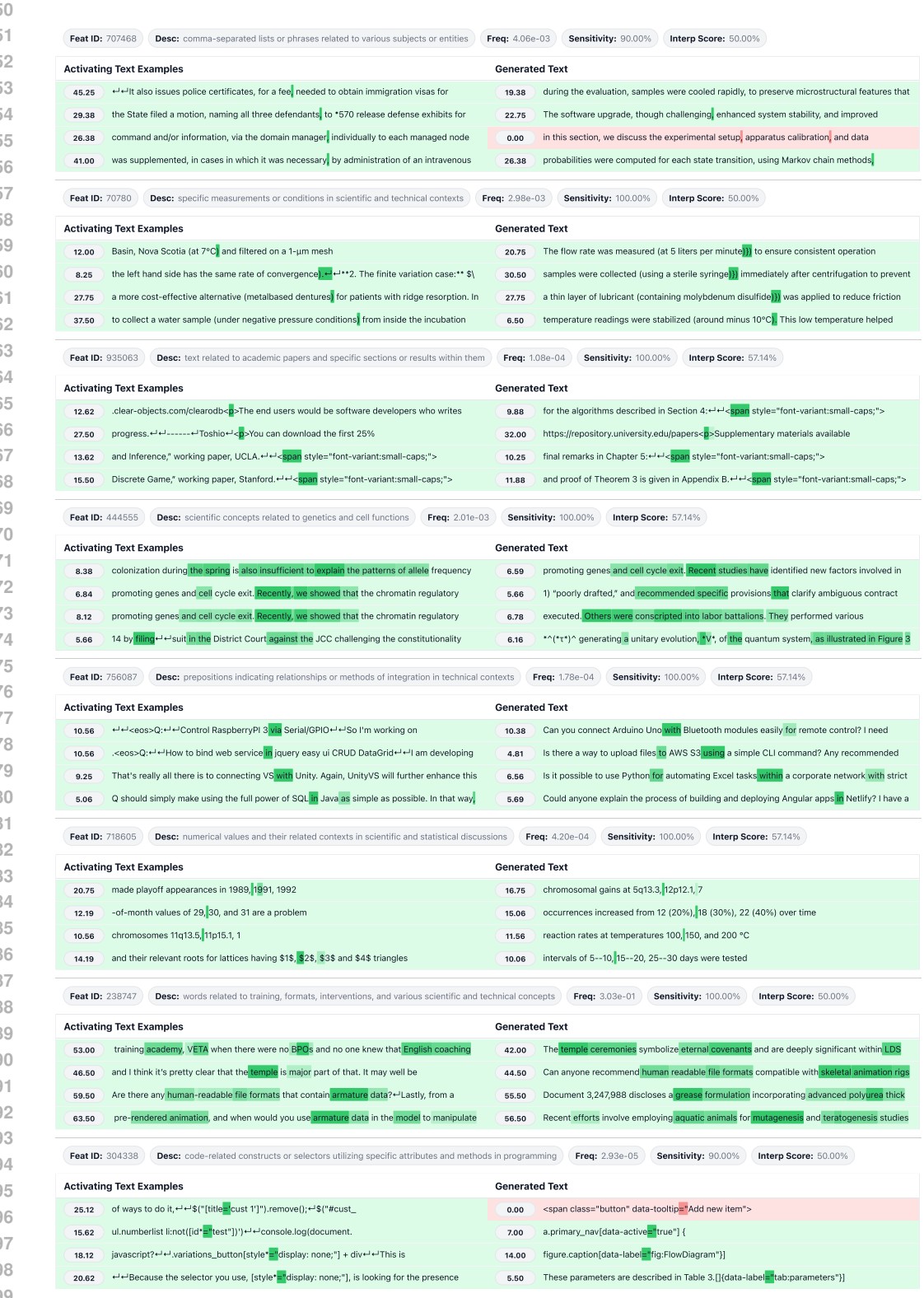

Figure 17: 8 randomly sampled features from those studied in Figure 3 that have high ($\geq 0.8$) sensitivity score and low auto-interp score ($\leq 0.6$). These features tend to be interpretable despite their low automated interpretability score.

