# OpenReview forum: "Measuring Sparse Autoencoder Feature Sensitivity"
_ICLR.cc/2026/Conference — Submitted to ICLR 2026_

### Official Review · Reviewer_unaA · 2025-10-30

**Soundness:** 3
**Presentation:** 3
**Contribution:** 3
**Rating:** 6
**Confidence:** 3

**Summary:**

This paper introduces a scalable method to evaluate SAE feature sensitivity—the probability that a feature activates on texts similar to its activating examples. The key innovation is using LLMs to generate semantically similar text without requiring natural language feature descriptions. The authors validate their approach through human evaluation and demonstrate that (1) many interpretable features exhibit poor sensitivity, and (2) average feature sensitivity declines with increasing SAE width across multiple architectures. This establishes feature sensitivity as a new dimension for evaluating both individual features and SAE architectures.

**Strengths:**

1. The explanation-free approach elegantly sidesteps the problem of imperfect feature descriptions that plague other interpretability evaluations
2. Human evaluation convincingly demonstrates that generated text genuinely resembles activating examples. The 79% "indistinguishable" rating provides strong support for the method's validity
3. Evaluation across 141 SAEs (29 GemmaScope + 112 SAEBench), spanning 7 architectures and multiple widths, provides robust evidence
4. The declining sensitivity with SAE width is an important discovery for the field, especially as it holds across architectures including Matryoshka SAEs designed to address scaling issues
5. Proper controls (frequency weighting), filtering justification, and statistical analysis strengthen conclusions

**Weaknesses:**

1. Filtering removes 35-79% of features (Table 2). While the 90% truncation threshold is justified, this substantially limits the generality of findings.
2. The 20-token total window (10 before + 10 after) may inadequately capture features requiring longer context. The paper acknowledges this but doesn't quantify the limitation
3. While documenting that sensitivity decreases with width is valuable, the paper provides limited insight into why: (1) Is this due to feature splitting/absorption? (2) Do wider SAEs learn more specific features that are harder to trigger? (3) What are the implications for practical use?

4. The binary activation criterion discards potentially valuable information about activation strength. Features might "weakly activate" on generated text, which could indicate partial sensitivity

**Questions:**

1. Have you analyzed how sensitivity varies with the typical context window required for feature activation? Can you categorize features by their context requirements?
2. Can you provide more analysis or hypotheses about why sensitivity declines with width? Does this relate to feature splitting, absorption, or other known SAE phenomena?
3. How do activation magnitudes on generated vs. activating text compare? Could weak activations indicate partial sensitivity?
4. Could you adapt your method for rare features by using the LLM to generate larger corpora of semantically similar text?

---

> ### Author Response · Authors · 2025-12-03
>
> We thank the reviewer for their careful reading and constructive feedback. We are glad you appreciated our explanation-free approach and comprehensive evaluation.
>
> **W1: High Filtering Rates**
>
> We agree that high filtering rates are a limitation, and we would be excited for future work to scale up activating text windows to reduce the need for filtering. In our top-level comment, we also introduce normalized sensitivity metrics (Appendix C) that avoid the need for filtering entirely and reproduce our main finding.
>
> **W2: Limited Context Window**
>
> We acknowledge the 20-token window may not capture features requiring longer context, though this follows prior autointerp settings. This limitation could be addressed in future work by scaling up the context window in our evaluation pipeline.
>
> **W3: Lack of Mechanistic Insight**
>
> We agree that understanding *why* sensitivity declines is an exciting direction for future work. Our contribution is establishing *that* this pattern holds robustly across 7 architectures.
>
> **W4: Binary Activation Criterion**
>
> We agree future work could consider activation activation strength. However, because features are so sparse, we expect binary activations to capture most of the effect. To our knowledge, how to best interpret activation magnitudes remains an open question for SAEs.
>
> **Q1: Sensitivity vs context window requirements?**
>
> Our current notion of sensitivity is most meaningful for features that reactivate within the context window. Characterizing features by their typical required context length is an interesting direction we are excited about.
>
> **Q2: Why does sensitivity decline?**
>
> We hypothesize this may be due to the same causes as absorption. Fundamentally, training SAEs on sparsity + reconstruction objectives does not incentivize features that are cleanly "canonical." However, we are excited for deeper investigation in future work.
>
> **Q3: Activation magnitudes?**
>
> There are likely more fine-grained approaches to measuring sensitivity that consider activation strength. However, because features are so sparse, we expect binary activations to capture most of the effect. In preliminary experiments, enforcing a minimum activation threshold did not meaningfully change sensitivity scores.
>
> **Q4: Adapting method for rare features?**
>
> Yes, we could directly adapt our method for lower frequency features by collecting activating examples over a larger corpus. The LLM generation step would actually have the same cost once activating texts are collected.

---

### Official Review · Reviewer_oWXM · 2025-10-30

**Soundness:** 3
**Presentation:** 3
**Contribution:** 2
**Rating:** 2
**Confidence:** 2

**Summary:**

The paper examines how SAEs can be quantitatively evaluated for interpretability and performance. It revisits metrics such as sparsity, reconstruction loss, and feature interpretability, highlighting trade-offs and limitations. The authors argue that current metrics fail to capture the nuanced balance between feature disentanglement and reconstruction quality. They propose improved metrics and evaluation setups that better reflect the practical goals of mechanistic interpretability.

**Strengths:**

1. The paper targets a crucial issue, how to measure SAE interpretability rigorously, an under-explored yet central topic in mechanistic interpretability research.

2. It systematically examines existing metrics (e.g., dead neuron rate, NMSE, feature overlap) and empirically demonstrates where they diverge from qualitative interpretability assessments.

3. Multiple model families (e.g., Pythia, Gemma) are analyzed. The paper reports trends across sparsity regimes and datasets rather than cherry-picking results.

**Weaknesses:**

1. The paper mostly refines and synthesizes existing metrics rather than introducing fundamentally new methods. Its contribution lies more in analysis than in methodological innovation.

2. While improved metrics are proposed, their theoretical justification is sometimes heuristic and lacks formal grounding or proof of desirable properties (e.g., invariance or sensitivity bounds).

3. The study shows correlations but doesn’t deeply explain why certain metrics diverge or what model mechanisms underlie these discrepancies.

4. There’s minimal comparison between metric-based interpretability and human judgments of monosemanticity or clarity, which would have strengthened validation.

**Questions:**

1. How robust are the proposed metrics to model size and dataset domain changes?

2. Can the metrics generalize to non-text modalities (e.g., vision, multimodal SAEs)?

3. How sensitive are metric rankings to the specific sparsity regularization used?

4. Do the proposed evaluation metrics correlate with human feature-interpretability scores?

---

> ### Author Response · Authors · 2025-12-03
>
> We thank the reviewer for their questions. We note that some aspects of the review may reflect a misunderstanding of our paper's specific contribution - our core contribute introduce a method for evaluating SAE feature *sensitivity* and studying sensitive at the feature and SAE level.
>
> We address the reviewer's concerns below:
>
> **W1: Limited Novelty**
>
> Our core contribution is a new pipeline for evaluating SAE feature sensitivity - measuring whether features activate on semantically similar text. Prior work studying feature sensitivity has needed expensive LLM data filtering, or been restricted to features that are predefined or can be modeled via n-grams. We discuss prior work in Section 2.1.
>
> **W2: Weak Theoretical Grounding**
>
> There is little precedent for theoretical analysis in automated interpretability evaluation. Formal guarantees would require restrictive assumptions about LLM sampling distributions. We instead rigorously validate our method empirically through human studies (Section 5.1) and automated tests of text diversity (Section 5.2).
>
> **W3: Lack of Mechanistic Insight**
>
> We agree that a deeper explanation of why sensitivity declines with SAE width is an exciting direction for future work.
>
> **W4: Missing Human Validation**
>
> We respectfully note that our paper includes human evaluation in Section 5.1. We conducted a blinded study where annotators rated 79% of our generated texts as "indistinguishable" from true activating examples (compared to 83% for ground truth). This validates that our method generates semantically appropriate text. Although we do not use human validation to check our interpretability metric, we use standard automated interpretability metrics that are widely accepted.
>
> **Q1: Robustness to model size and dataset domain?**
>
> We evaluate across two models (Pythia 140m and Gemma2 2B), finding consistent scaling trends. Each SAE is inherently tied to its base model and training data, so our evaluation metric by definition depends on the model and dataset used to train a given SAE.
>
> **Q2: Generalization to non-text modalities?**
>
> Our method requires an LLM to generate semantically similar examples, so adaptation to vision would require a different generation approach. However, the concept of feature sensitivity generalizes naturally to other modalities.
>
> **Q3: Sensitivity to sparsity regularization?**
>
> We find that SAEs with more active latents have higher sensitivity. However, our core finding that wider SAEs have declining sensitivity holds across all sparsity levels. Figure 6 shows the interaction between sparsity, dictionary size, and sensitivity.
>
> **Q4: Correlation with human interpretability scores?**
>
> One of our core results is that automated interpretability and sensitivity often disagree (Figure 3). Upon human inspection, we found that when features have high sensitivity but low interpretability, this is usually because the autointerp is noisy and the feature is actually interpretable. However, we also confirm via a human study that features with high interpretability and low sensitivity are real.

---

### Official Review · Reviewer_oHCF · 2025-11-01

**Soundness:** 3
**Presentation:** 2
**Contribution:** 2
**Rating:** 4
**Confidence:** 2

**Summary:**

This paper introduces "feature sensitivity," a novel metric for evaluating Sparse Autoencoder (SAE) features that measures a feature's ability to activate on semantically similar inputs, not just its canonical examples. The authors propose a clever pipeline that uses an LLM to generate new test cases directly from a feature's activating examples. The work's central findings are that many highly interpretable features surprisingly exhibit poor sensitivity and, more critically, that average feature sensitivity consistently declines as SAE width increases. This result is presented as a new, fundamental challenge for scaling SAEs.

**Strengths:**

1. The paper identifies and formalizes a critical gap in SAE evaluation. The distinction between interpretability (cohesive activating examples) and sensitivity (robust generalization) is a valuable conceptual contribution that pushes the field toward a more rigorous understanding of feature quality.

2. The authors perform solid validation of their method, including a blinded human evaluation that confirms the quality of the generated text and an automated analysis of text novelty.

**Weaknesses:**

1. The paper identifies that features have varying sensitivity but provides no actionable insights: What should practitioners do with features that have low sensitivity? There is no evidence that sensitivity (as defined here) correlates with utility for downstream tasks (steering, circuit analysis, unlearning, etc.). No solutions or training modifications are proposed to improve sensitivity.

2. The paper filters out 35-79% of features, with exclusion rates dramatically increasing with SAE width (51% for 65K to 79% for 1M). The main finding (wider SAEs → lower sensitivity) could be entirely explained by differential filtering rates rather than true sensitivity differences. Filtering for features that "activate on their own truncated examples" explicitly selects for simpler, context-independent features, which is acknowledged but not adequately addressed. Table 2 shows that this filtering becomes more severe precisely where the sensitivity decline is observed, creating a plausible confound. The robustness checks in Figure 9 don't fully resolve this since different cutoffs still show the same filtering pattern.

**Questions:**

Please see the weaknesses.

---

> ### Author Response · Authors · 2025-12-03
>
> We thank the reviewer for their thoughtful engagement and for identifying utility and filtering as key concerns.
>
> **W1: Actionable Insights / Downstream Utility**
>
> We agree that our work would benefit from a greater discussion of downstream implications. In our top-level comment, we present preliminary results showing that sensitivity correlates with performance on downstream tasks (Appendix D). Of 5 downstream tasks from SAEBench, 4 show significant correlations with sensitivity (3 positive, 1 negative).
>
> **W2: Filtering Confound**
>
> We agree that filtering exclusion rates are surprisingly high and that this is a limitation. However, we note that the expected impact of filtering is to *increase* measured sensitivity. Stricter filtering results in higher sensitivity across the board (Figure 9). This makes our result more surprising: despite filtering out more features for wider SAEs (which should artificially inflate their measured sensitivity), we still observe declining sensitivity.
>
> To further address this concern, we introduce two normalized metrics in Appendix C that account for varying truncated activation rates *without any filtering*. Using the sensitivity ratio and sensitivity gap averaged across all GemmaScope features, wider SAEs still show lower sensitivity, reproducing our main finding.
>
> We hope this addresses the reviewer's concerns and welcome further discussion.
> is

---

### Official Review · Reviewer_Ztja · 2025-11-01

**Soundness:** 2
**Presentation:** 2
**Contribution:** 2
**Rating:** 2
**Confidence:** 4

**Summary:**

Sparse Autoencoders are a very popular method to decompose LLM activation space into interpretable features but because they are trained in an unsupervised way we don't have ground truth features. As a result, evaluation of feature quality is difficult. The authors propose a new method to measure one specific aspect that makes a good feature: Sensitivity, i.e. whether the feature activates for prompts semantically similar to activating examples.

The major flaw in their approach is the assumption that the LLM samples **should** activate the feature. This is not true in practice, and as a result, their metric measures either whether the feature is sensitive, or whether the generator LLM's implicit feature description is good and we cannot distinguish both, thereby limiting the usefulness of this metric.

**Strengths:**

- the research question (how to evaluate SAE feature quality) is very important given SAE's popularity and also neglected
- the experiments done are sound and mostly well executed

**Weaknesses:**

The central assumption that the entire method relies on is that LLM-generated, semantically similar samples should activate the feature. Why should this assumption hold? The authors don't present any idea or empirical data and simply assume this is true. But it is important: If the LLM generates samples where the features should NOT activate (examples below), then the sensitivity score could be entirely wrong.

The authors present not needing a feature annotation as a main advantage of their strategy. However, although they don't require an **explicit** annotation, the LLM sample generator needs to make an **implicit** guess about what the feature might encode, in order to generate activating examples. If this guess is wrong, the LLM would generate samples that should not activate and the sensitivity score is low not because the SAE feature is low-quality, but because the LLM doesn't understand what makes the shown activating examples really activate.

To illustrate why I expect this problem to be the **norm** and not an edge case, I'd like to give some examples.

Example 1: Ground truth feature the SAE found is "The subject of this sentence is the second noun", and the feature fires at positions that predict a noun.
This is a feature that should exist in real language models as it's part of the IOI circuit (Wang, 2022) and real SAE features have actually been found (Makelov, 2025). So this is a feature that is great, monosemantic, with a clear interpretation, and many such features are present in real SAEs.
Let's pretend this SAE feature is perfect, ie sensitivity=100% and think through how high the proposed method would score the sensitivity. Input prompts with activating examples would be seemingly arbitrary text, with many activations being at tokens like "to", "the", "for", etc, since these words often predict a noun that should possibly be copied from earlier in the context. The LLM sample generator would now just generate arbitrary sentences that just so happen to contain those tokens but no specific sentences where the subject/object should be copied and thus the proposed sensitivity metric would be low although the "ground truth" sensitivity is high.

Example 2: Features, that are better explained by their role on the LLM output. An LLM has to understand input but also generate output and many features, especially in late layers have shown to be better explained by their role in crafting the model's prediction. Often, activating this neuron leads to promoting or inhibiting a meaningful set of tokens as has been shown by simply calculating f @ W_dec @ W_unemb and most features in late layers behave like this. For example, let's assume a feature is active whenever the model predicts a form of the word "sell". This feature could be identified quite easily by calculating f @ W_dec @ W_unemb and showing that activating it leads to the logits of [" sold", "sold", " sell", " resold", ...] being higher.
Here's the link for this feature on neuronpedia: https://www.neuronpedia.org/gemma-2-2b/19-clt-hp/40860
Looking at activations alone, it activates on words like "which", "it", "to", for" in various different contexts. Only a small subset of prompts actually contain a form of the word sell at the next token position, because often, the model's prediction is different than the actual text. Thus, again, the LLM sample generator would generate samples where most don't activate the feature because the LLM has no idea what the feature really activates for. So sensitivity would be low although the feature learned is perfect.

This problem gets worse with scale:
- larger models have more layers and CoT tokens, so most features need to be explained in terms of their interactions with other features rather than inputs.
- more intelligent models might have more abstract, complex features, still monosemantic, but in a nontrivial way. Simply generating some sentences that appear similar wouldn't cut it.

As a result, the proposed metric is not a principled method to evaluate SAE feature quality because it cannot distinguish whether the LLM sampler understands the nature of this feature vs the feature itself has low sensitivity.

**Questions:**

- methodological details are missing for the human study: number of human evaluators, who they were (e.g. MI researchers, undergraduates, etc), how many prompts did each person score, how was blinding done, etc.
- ll 448 mentions results for Matryoshka SAEs but no data or figure is linked. Where can I find the results of this experiment?
- Figure 3b, could the authors clarify which Auto-Interp Score was calculated here? (e.g. detection, fuzzing, simulation, something else)
- Figure 3b, c, d, why are the scores clustered around decimals?
- Only 2M tokens were used to produce activating examples. Since many features occur rarely (ie frequency 1e-5 or less), I expect that many features only have few activating examples. The authors acknowledge this limitation but there's a deeper flaw: Because you sample so few, you likely miss the highest activating, often most clear and interesting examples. If more activations were used, the activating examples might have been clearer, the LLM would have been able to generate better samples, and the sensitivity score could've been possibly much higher.

---

> ### Author Response · Authors · 2025-11-24
>
> We thank the reviewer for their engagement with our paper. We are glad the reviewer found SAE sensitivity to be important and neglected research question .
>
> We fully agree with the assessment that a load bearing piece of our method is the assumption that our LLM samples should activate the SAE feature.
>
> **We directly test this concern in Section 5.1 using a blinded human study.** We focused on interpretable features where text generated by our method failed to activate the feature. We showed these texts to blinded human annotators alongside activating examples and asked them to rate their relatedness. We found that 79% of our generated texts were rated "indistinguishable" compared to 83% for true activating examples. Generated text achieves relevance ratings nearly matching ground truth, confirming that low sensitivity evaluations reflect poor sensitivity rather than poor generation.
>
> The key point we want to convey is that our generator can match deep semantic properties rather than just shallow word matching. We appreciate the concrete examples provideded as stress tests. **We ran our exact generation pipeline on both examples present results below.** In text below, {{}} denote either feature activations in example text, or where the generator expects the feature to activate (these are stripped during testing). Our method produced 10 samples and we show the first 3 for each case without cherry-picking.
>
> *Example 1: IOI Subject Tracking Feature*: We were unable to locate the SAEs used in Makelov et al. 2025, so we instead hand-wrote 15 activating text examples following the IOI structure where the second name is the subject, e.g. "Mary and John went to the store. It was crowded that day. John gave the book {{to}} Mary."
>
> Generated samples:
> - As soon as Eric and Fiona arrived at the theater, Fiona handed the tickets {{to}} Eric.
> - While David and Sarah were hiking through the forest, Sarah pointed {{at}} a deer that crossed their path.
> - During the meeting, Alex and Christina reviewed the budget proposal, and Christina explained the figures {{to}} Alex.
>
> *Example 2: "Predict sell" feature*: We used the activating examples directly from the Neuronpedia link provided.
>
> Generated samples:
> - The latest report from the estate shows the rare collection of antique furniture {{was}} finally auctioned off for a staggering sum.
> - The committee decided {{to}} implement new guidelines that would enhance the efficiency of the project management process moving forward.
> - A detailed analysis of the data presented in the study shows significant improvements in overall performance {{when}} adopting the new strategies proposed.
>
> For both features, generations successfully capture the underlying semantic patterns rather than surface-level token matching.
>
> **We lastly answer the reviewer's specific questions:**
>
> **Q:** *"methodological details are missing for the human study: number of human evaluators, who they were, how many prompts did each person score, how was blinding done"*
>
> **A:** Two authors served as annotators, each scoring approximately half the prompts. We randomized presentation order without revealing sample categories to annotators.
>
> **Q:** *"ll 448 mentions results for Matryoshka SAEs but no data or figure is linked"*
>
> **A:** Matryoshka results appear in Figure 7 (sensitivity vs sparsity) and Figure 8 (sensitivity vs dictionary size), shown as red diamonds in left subplots.
>
> **Q:** *"Figure 3b, could the authors clarify which Auto-Interp Score was calculated here?"*
>
> **A:** We use the detection score as implemented in SAEBench [Karvonen et al. 2025].
>
> **Q:** *"Figure 3b, c, d, why are the scores clustered around decimals?"*
>
> **A:** Our sensitivity metric is discrete (fraction of 10 samples that reactivate the feature). We add small y-axis jitter for visualization purposes.
>
> **Q:** *"Only 2M tokens were used... you likely miss the highest activating examples. If more activations were used, the sensitivity score could've been possibly much higher."*
>
> **A:** We agree collecting text for more tokens could yield clearer examples. However, Figure 3c shows no significant positive correlation between feature frequency and sensitivity, confirming our metric is not conflated with feature frequency.
>
> We appreciate the reviewer's identification that generation test sample quality as the crux of our method.  We hope the human evaluation in Section 5.1 validation and these additional results on the provided example features address this core concern. We would welcome the reviewer's thoughts on whether this evidence updates their assessment and are happy to answer any further questions.

---

### Author Response · Authors · 2025-12-03
**General response and new changes**

We thank all reviewers for their feedback and engagement. We are glad reviewers appreciated the importance of this work: Ztja found "the research question (how to evaluate SAE feature quality) is very important given SAE's popularity and also neglected," oHCF writes that we "identify and formalize a critical gap in SAE evaluation" and that "the distinction between interpretability and sensitivity [is] a valuable conceptual contribution," and unaA describes our finding that sensitivity declines with SAE width as "an important discovery for the field."

We address two common concerns below with **two new analyses added to the appendix**: (1) correlation between sensitivity and downstream task performance (Appendix D), and (2) normalized sensitivity metrics that require no filtering (Appendix C).

**1. Downstream task correlation**

Several reviewers asked about practical utility of our sensitivity metric. As a first step, we conducted a preliminary analysis examining the relationship between sensitivity and downstream task performance across 42 SAEs trained on Gemma-2-2B from SAEBench. These SAEs span multiple architectures and sparsity levels. Sensitivity shows significant positive correlations with SCR (r = 0.43, p = 0.005), TPP (r = 0.33, p = 0.036), and RAVEL (r = 0.43, p = 0.005), and a significant negative correlation with Unlearning (r = -0.37, p = 0.017).

We emphasize that these results are exploratory: we do not control for other SAE attributes (e.g., L0, reconstruction quality) or correct for multiple comparisons. However, we present this as preliminary evidence that sensitivity correlates with downstream performance. Full results are in Appendix D.

**2. Normalized sensitivity without filtering**

Reviewers raised concerns that our filtering approach (which excludes 35–79% of features) may confound our scaling results. To address this, we introduce two normalized metrics that account for varying truncated activation rates without any filtering.

We define the sensitivity ratio as min(q/p, 1), measuring what fraction of a feature's truncated-text activation rate is achieved on generated text. We define the sensitivity gap as min(p - q, 0), measuring the drop in activation rate. Averaging these metrics across all GemmaScope features (with no filtering), wider SAEs still show lower sensitivity ratios and larger sensitivity gaps, reproducing our main finding. Full results are in Appendix C.

---

### Meta-Review · Area_Chair_th5h · 2026-01-07

**Summary:**

This work takes a step towards using feature sensitivity as a new dimension for evaluating both individual features and sparse autoencoder architectures. The authors develop a pipeline that generates texts similar to SAE feature activating examples which they also validate with human judges.

The main concerns raised by the reviewers are the following: 1/ unclear relation between  feature sensitivity and downstream task performance; 2/ possible confounding effect of filtering at evaluation time.

Additional concerns raised by individual reviewers:
- Ztja: Assumption that LLM samples should activate the SAE feature.
- unaA: Features requiring longer context and various interpretation questions.

**Reviewer Concerns:**

The authors addressed the two main concerns by providing two new analyses added to the appendix:

1/ Correlation between sensitivity and downstream task performance. The results produced by the authors, who qualified them as preliminary, only exhibit modest correlations between these quantities. Hence, this analysis shows that the concern raised by the reviewers is valid and only partially resolves the concern.

2/ Normalised sensitivity metrics without filtering. The authors introduced two normalised metrics that account for varying truncated activation rates without any filtering. As far as I am concerned, these additional results resolve the issue and provide the supporting evidence asked by the reviewers.

The authors provided detailed responses to most of the concerns raised by the reviewers, but remained relatively shallow regarding insights and interpretations of why the proposed method works.

**Reviewer Scores:**

Most reviewers voted for rejection with the better scores exhibiting relatively low confidence. Given the additional supporting evidence provided by the authors I would have expected the reviewers to stick to their scores given the modest new insights offered by authors, making this a borderline reject paper.

---

### Decision · Program_Chairs · 2026-01-26

Reject